# Correlation Between Neurocognitive Outcomes and Neuroaxonal Connectome Alterations After Whole Brain Radiotherapy: A Proof-of-Concept Study [note 1]

**DOI:** 10.3390/cancers17111752

**Published:** 2025-05-23

**Authors:** Sreenija Yarlagadda, Starlie Belnap, John Candela, Tugce Kutuk, Thailin Companioni Reyes, Miguel Ramirez Menendez, Matthew Hall, Robert Press, Yazmin Odia, Minesh Mehta, Michael McDermott, Rupesh Kotecha

**Affiliations:** 1Miami Cancer Institute, Baptist Health South Florida, Miami, FL 33176, USA; sreenija.yarlagadda@baptisthealth.net (S.Y.); tugcek@baptisthealth.net (T.K.); thailin.companionireyes@baptisthealth.net (T.C.R.); miguelar@baptisthealth.net (M.R.M.); matthewha@baptisthealth.net (M.H.); robert.press@baptisthealth.net (R.P.); yazmino@baptisthealth.net (Y.O.); mineshm@baptisthealth.net (M.M.); 2Miami Neuroscience Institute, Baptist Health South Florida, Miami, FL 33176, USA; starlieb@baptisthealth.net (S.B.); john.candela@baptisthealth.net (J.C.); mwmcd@baptisthealth.net (M.M.); 3Herbert Wertheim College of Medicine, Florida International University, Miami, FL 33199, USA

**Keywords:** connectomics, brain networks, brain mapping, neurocognitive outcomes, magnetic resonance, diffusion tensor imaging, whole brain radiotherapy

## Abstract

Brain connectivity and its modulation by tumor-related treatment has garnered recent interest with developments in the field of “Connectomics”, which involves integrating data from multiple advanced imaging modalities and mapping the complex brain networks, providing insights into brain disorders. In the current study, we aimed to evaluate the effect of whole brain radiotherapy (WBRT) in patients with brain metastases on human brain networks and neurocognitive outcomes. In a prospective registry, data was integrated from diffusion tensor imaging and functional magnetic resonance imaging, and individualized brain maps of 15 neuronal networks were created for each patient at different time points. We observed that the baseline anomalies due to brain metastases increased post-WBRT across multiple networks. The anomaly changes also correlated with neurocognitive decline, suggesting the importance of functional connectivity in cognitive processes. The long-term outcomes might suggest connectome-tailored radiation dose constraints to preserve cognition.

## 1. Introduction

The collaborative work between neuroscience and network science to characterize complex brain functionality led to a new branch called “Connectomics” that evaluates the structural and functional connectivity of the brain [1]. Diffusion tensor imaging (DTI) with tractography explores the structural connectivity of the brain and provides a detailed model of the orientation and integrity of white matter fiber bundles [2,3]. Resting-state functional magnetic resonance imaging (*f*MRI) evaluates the biologically plausible functional areas of the brain and measures temporal correlations between them, providing insights into the intrinsic functional architecture and connectivity patterns in various neurological conditions, including aberrations that arise in these conditions [4,5,6,7,8]. Advances in modern brain mapping techniques, including complex network analysis, integrate data from these modalities, yielding “human brain networks” [9,10,11]. To understand the brain–cognition–behavior relationship, imaging-based “parcellations” and “nodes” were defined in a network, with the functional connections termed “edges” [12,13].

The Human Connectome Project (HCP) is a large-scale research initiative that aims to map brain connectivity in a comprehensive healthy population and provide an open-access database of imaging and behavioral data [14,15]. Research using the HCP led to the recognition of a fingerprint to represent each cortical area and the creation of the first multi-model atlas of the human brain [16]. This research has been applied to guide connectome-based resections to reduce the incidence of surgery-induced neurological deficits while maintaining onco-functional balance [17,18,19]. Additionally, preliminary studies on patients with primary brain tumors receiving radiation therapy (RT) demonstrated alterations in network topology [20,21,22]. Whether these changes contribute directly to cognitive decline is yet to be demonstrated.

A prospective imaging registry trial (IRB: BHSF1701633) was initiated at our institution to develop a longitudinal imaging repository facilitating translational research. The current analysis expands on the study presented in [23,24] with the objective to evaluate the effects of whole brain radiation therapy (WBRT) on the human connectome and correlate these with changes in cognition. In this proof-of-concept study, we report the imaging findings of the first 10 patients enrolled in the registry trial and correlate the changes in connectomes with alteration in neurocognitive function.

## 2. Materials and Methods

Patients with brain metastases treated with WBRT were enrolled in the prospective imaging registry trial. English- or Spanish-speaking patients who consented to undergo a minimum of two magnetic resonance imaging (MRI) sequences were included.

Figure 1 depicts the workflow for the creation of brain networks. After obtaining informed consent, in addition to the standard-of-care MRI (including at least an Axial T1 weighted sequence with contrast), DTI and *f*MRI (resting state-blood oxygen level dependent) were acquired for each patient at each specific time point [25]. The DICOM files were uploaded to Infinitome (Omniscient Neurotechnology Pty Ltd., Sydney, Australia), a cloud-driven, HIPAA-compliant research platform where the diffusion-weighted images (DWI) were corrected for motion and gradient artifacts and co-registered to the T1 weighted sequence; the structural connectivity of white matter fibers was derived using Tournier’s method [26]. Quicktome Discovery mode™ AI brain mapping software (https://www.o8t.com/quicktome (accessed on 20 May 2025)) was used that incorporates a machine learning model trained on neuroimaging data from 178 healthy individuals obtained from Schizconnect (http://schizconnect.org) and 40 brain tumor patients enabling accurate parcellation on structurally distorted brain parenchyma and used gradient-boosted decision tree model (XGBoost) to build a statistical map between voxel level feature vector and most probable parcellation class [27]. A subject-specific HCP multi-model parcellation version 1 atlas with 15 neuronal networks for each patient at each specific time point was created. These networks are detailed in Table 1. These brain maps were then compared to normal brains from the HCP, and a connectivity-based anomaly matrix was created (Figure 2) for 379 parcellations (Quicktome Discovery mode™). A 3 standard deviation alteration from the normal brain was defined as an “anomaly”. Hyperactivation was defined as the area generating more neuronal activity compared to normal and hypoactivation when showing depressed neuronal activity. Connectome analysis was performed at baseline (pre-WBRT) and 3 months post-WBRT using the same image capturing technique, reconstruction, and processing software to ensure uniformity. The 3-month time interval for follow-up scan was chosen given the prior data on the timing of neurocognitive change after WBRT [28,29].

As a pragmatic alternative to the traditional resource-intensive, paper-and-pencil battery-based neurocognitive assessment, a prospective clinical trial was also initiated at our institute (NCT05504681) using a novel, interactive, multi-dimensional, app-based assessment battery called “Cognition” (Brainlab AG, Munich, Germany), which was developed to evaluate patients with brain metastases [56,57]. Neurocognitive assessment included tests for the five domains: (1) learning and memory, (2) attention and speed of processing, (3) verbal fluency, (4) fine motor and speed, and (5) executive functions (details included in Table 2). A comprehensive description of the app-based testing was reported in our earlier study [58]. The patients underwent these neurocognitive evaluations prior to treatment and at 3 months after WBRT.

Anomaly scores were captured at baseline and post-WBRT. Descriptive statistics were used to define patient baseline characteristics and compared to the anomaly frequencies in each network between baseline and post-WBRT. Neurocognitive function change was computed using the Iverson modification of Reliable Change Index (RCI), and in accordance with the previous literature, a *z*-score of ±1.645 (90% confidence level) was defined as reliable change [59]. Pearson correlation was used to determine the relationship between anomaly frequency and neurocognitive decline and presented as Pearson coefficient (r). Statistical analysis was performed using SPSS version 27 (SPSS Inc., Chicago, IL, USA).

## 3. Results

### 3.1. Baseline Characteristics of Study Population

Ten consecutive patients who were treated with WBRT and underwent connectome analysis pre- and post-WBRT were included in this proof-of-concept analysis. The median age was 62 years (range: 52–79 years), the majority were female (70%), and the median Karnofsky performance score (KPS) was 90% (range: 70–90%). Lung (*n* = 7) was the most common primary site of disease followed by breast (*n* = 2) and renal cell carcinoma (*n* = 1). Half (5/10) of the patients were treated with hippocampal-avoidant WBRT, and the other half were treated with conventional WBRT to a median dose of 30 Gy in 10 fractions. Two patients had prior resection of brain metastases, and four had prior stereotactic radiosurgery (22–24 Gy in single fraction). The median duration between prior intra-cranial treatment and WBRT was 7 months. Post-WBRT, patients received targeted therapy (*n* = 4), immunotherapy (*n* = 3), or chemotherapy (*n* = 1). The median duration between WBRT completion and the start of systemic therapy was two weeks.

### 3.2. Baseline (Pre-WBRT) Connectome Evaluation

At baseline, prior to WBRT, connectome analysis revealed significant alterations in functional connectivity across multiple networks. With a median of 10 (range: 8–14) abnormal networks per patient, the highest proportion of anomalies per network (average across all patients) was noted in the multiple demand network (MDE: 46%), followed by the paralimbic network (LIMPA: 31.5%) and the central executive network (CEN: 30.8%). Among the parcellations, p10p (posterior 10 polar, which is a part of lateral frontal lobe regions involved in episodic and working memory tasks) had the highest anomaly frequency of 15, followed by 14 in te1p (temporal area 1 posterior, part of temporal lobe, primarily related to visual pathways and visual working memory) and s6–8 (superior 6–8, part of lateral temporal lobe, involved in maintenance of spatial information).

### 3.3. Post-WBRT Comparison

Post-WBRT connectome analysis revealed increases in the anomaly frequencies in 8 networks, with a median of 12 (range: 6–14) abnormal networks per patient; the highest proportion of anomalies per network was in the LIMPA (47.3%), followed by the MDE (43.2%) and the CEN (34.2%). Figure 3 depicts a representative case showing alterations in connectivity post-WBRT. The highest proportional increase in anomaly frequency was observed in the limbic network (LIM: 73%), followed by LIMPA (50%), and a proportional decrease in the anomaly frequency in the default mode network (DMN: 55%), followed by visual network (VIS: 37%) (Table 3). Parcellations ip1 (intraparietal 1: part of lateral parietal lobe, involved in mental arithmetic activities), scef (supplementary and cingulate eye field: part of medial superior frontal gyrus, involved in goal-directed behavior), p9–46v (posterior 9–46 ventral: part of lateral frontal region, involved in goal-directed and higher-order cognitive processes) had the highest anomaly frequency of 14. The patients who received any prior intracranial treatment (SRS or resection) exhibited a substantial increase in the total anomaly frequency of 53% (across all networks), while those with no prior treatment had a decrease of 33%.

### 3.4. Neurocognitive Outcomes

The neurocognitive assessments conducted at similar intervals in six patients demonstrated a decline in learning and memory: verbal recall in 33%, verbal revision in 50%, verbal recognition in 50%, attention and speed of processing in 33%, verbal fluency in 50%, and fine motor and speed in 17% of patients compared to their own baseline assessment. Pearson correlation showed a very strong correlation between neurocognitive domain decline and anomaly changes: learning and memory domain with SCN [Verbal recall (Pearson coefficient −0.94; *p* < 0.01), verbal revision (Pearson coefficient −0.89; *p* = 0.01), and verbal recognition (Pearson coefficient −0.94; *p* < 0.01)] (Figure 4). The proportional anomaly frequency in SCN decreased by 11% in hippocampal avoidant WBRT (*n* = 5), while it increased by 133% in conventional WBRT (*n* = 5).

## 4. Discussion

To the best of our knowledge, this is the first study to explore the effect of WBRT on the connectome and characterize the change in anomaly frequency. Although this is a preliminary evaluation, we found that changes in the SCN correlated with a decline in learning and memory, and intriguingly, SCN anomaly frequency differed between hippocampal avoidant and conventional WBRT. Collectively, these findings suggest the relevance of brain network connectivity in cognitive functioning, which may contribute to connectome-based radiation dose constraints in the future.

In our study, network anomaly reflects hyper- or hypo-activation relative to healthy controls from the HCP. However, in this analysis, the post-WBRT anomaly frequency was compared to the patient’s own baseline value, considering that tumor-specific alterations were already abundant at baseline evaluation. During the follow-up assessment, none of the patients had disease progression in the brain, indicating the anomaly changes and cognitive decline observed during this period could not be attributed to tumor-related changes.

The key components of SCN include the intricate connections between basal ganglia, thalamus, hypothalamus, and hippocampus. Traditionally, these structures were studied as isolated anatomic loci; however, their reconceptualization as networks with coordinated neuronal activity and dynamic hubs contributing to critical cognitive functions is yielding significant clinical implications. Learning and memory processes rely extensively on the dynamic interaction between cortical and subcortical structures, with differential activity across memory encoding, consolidation, and retrieval phases [60,61]. Thalamic sub-nuclei are well known to form functional circuits with extensive reciprocal connections to cortical areas, and act as a dynamic filter of information flow. The mediodorsal thalamus enhances fronto-parietal interactions, stabilizing information during the “verbal recall” phase [62]. Concurrently, the basal ganglia reinforces verbal associations via dopaminergic signaling and is central to procedural learning [63]. Thus, “verbal revision” and “verbal recognition” require coordinated thalamocortical and basal ganglia–cortical interactions. Xu et al. compared individuals with subjective cognitive decline to healthy controls and reported significant structural and functional alterations in the cortical–subcortical circuit and the disappearance of some hubs [64]. The hippocampus binds multimodal sensory, temporal, and contextual details into coherent memories. It interacts extensively with the medial prefrontal cortex (mPFC) and plays a key role in episodic and spatial memory [65]. Dynamic causal modeling showed effective connectivity between the hippocampus and DMN during encoding and memory retrieval [66]. Although a statistical difference was not attained due to the small sample size of the current study, the difference observed in the change in SCN anomaly frequency between hippocampal avoidant and conventional WBRT reinforces the effect of RT on brain networks and might indicate a threshold dose responsible for anomalies that attain clinical significance.

Bahrami et al. previously studied the effect of RT on structural connectivity and found increased modularity along with increases in transitivity and suggested poorer communication across modules and networks after RT [20]. Sleurs et al. evaluated white matter microstructural alterations in children who received RT for infratentorial tumors and demonstrated widespread lower fiber density across the brain and lower fiber cross-section at the treated areas [22]. They also observed lower global and local efficiency across all the network costs compared to controls and identified hubs (most densely connected areas) were most significantly impacted, and demonstrated a strong correlation with intelligence scores. They suggest that sparing these hubs during RT/surgery might be of value to preserve long-term cognition. Our study analyzing both structural and functional connectivity concurs with the findings of these two studies in terms of worsening in the anomaly frequencies post-RT. The presence of brain tumors, per se, causes topological alterations in the networks, seemingly further increased by RT.

Sporns et al. suggest that brain networks demonstrate interlinked communities that form a partly decomposable modular architecture, and such architecture is of fundamental importance for understanding mental processing and cognition [67]. The ability of the brain networks to reconfigure dynamically, adapting to the environment, has been studied. A prior study reviewing functional connectomes in a longitudinal cohort of six glioblastoma patients showed a transient deterioration in network anomalies, followed by near-total recovery, 2 months post-surgery [68]. While our current study is limited by its sample size, which may affect the generalizability of results, we plan to investigate the trends in network alterations with reconfigurations over time in a larger cohort with a longer follow-up. Additionally in this study, we used a novel neurocognitive testing app in the prospective cohort to evaluate neurocognitive changes across multiple domains, which in earlier publications demonstrated results comparable to traditional paper-and-pencil battery tests [57]. This is a limitation of our findings, as the novel test employed has not been validated across diverse populations.

The current study followed a rigorous workflow constituted by several phases, specifically image acquisition, data pre-processing, brain parcellation, and anomaly matrix generation for each network at each specific time point. While the correlation between changes in anomaly frequency and neurocognitive decline is determined, the pathophysiology of radiation-induced altered connectivity has yet to be explored. As our study suggests, a threshold dose can cause the anomalies to attain clinical significance or can result from a stochastic effect, which is yet to be determined. Future efforts in this direction can yield a connectome-tailored treatment directive.

## 5. Conclusions

This study represents the first evaluation of the human connectome before and after WBRT. Integration of the data from DTI and *f*MRI can facilitate the creation of human brain networks, which reveal significant changes in functional networks, parcellations, nodes, and hubs that are not appreciated by standard follow-up MRI and correlate with neurocognitive testing on preliminary analysis. Further studies in a larger cohort are underway, and correlations with neurocognitive outcomes, white matter changes, and tumor locations/numbers will be performed.

## Figures and Tables

**Figure 1 cancers-17-01752-f001:**
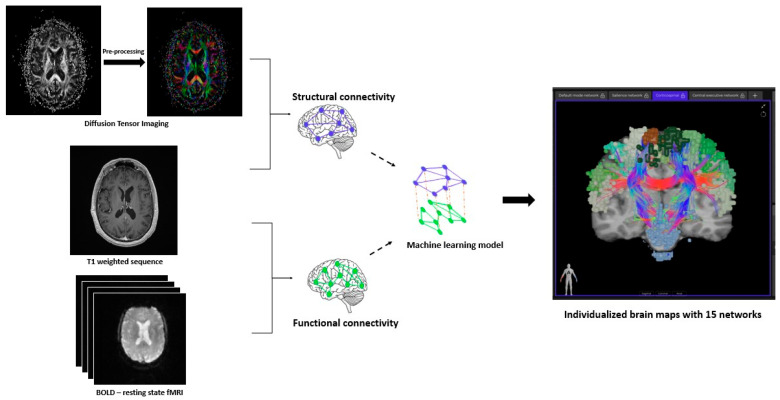
Workflow for the creation of brain networks. The diffusion-weighted images (DWI) were pre-processed by correcting for motion and gradient artifacts and co-registered to the T1 weighted sequence to analyze the structural connectivity of white matter fibers. A resting-state blood oxygen level dependent (BOLD) functional MRI (fMRI) co-registered to the T1 weighted sequence gives the functional connectivity of brain. A machine learning model integrates data from these modalities and creates subject-specific neuronal networks (15) for each patient.

**Figure 2 cancers-17-01752-f002:**
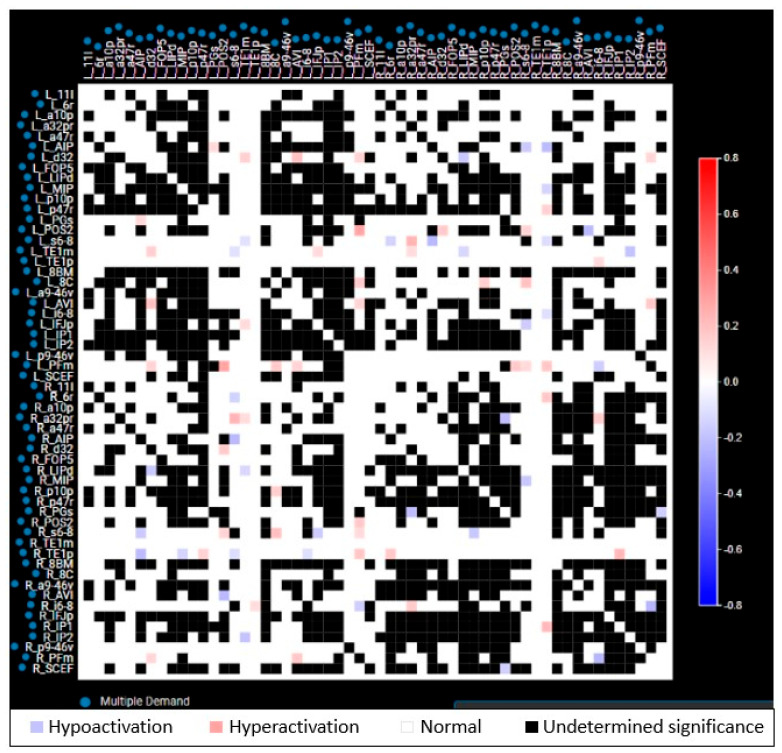
Representative anomaly matrix of a single network (multiple demand network) in a patient. Each parcellation of a network is compared to the respective parcellation of template normal brain network from Human Connectome Project (HCP), and the functional status is depicted as blue (hypoactivation), red (hyperactivation), white (normal), and black (undetermined significance). Both red and blue are considered as anomalies.

**Figure 3 cancers-17-01752-f003:**
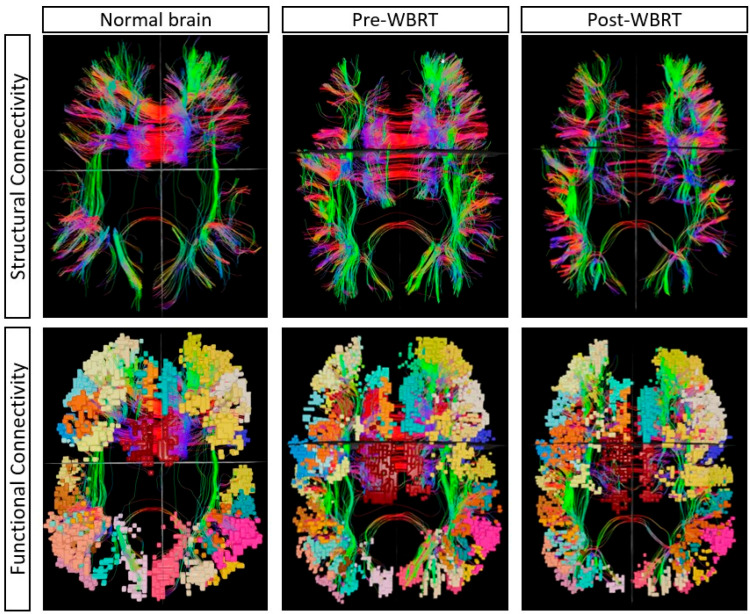
A single representative case showing the changes in multiple demand network pre- and post-WBRT in comparison to a normal brain from the Human Connectome Project. The alteration in structural connectivity is visualized with decreased fiber density at baseline (pre-WBRT), which further deteriorated post-WBRT. The anomaly frequency at baseline (pre-WBRT) was 13, which increased to 32 post-WBRT.

**Figure 4 cancers-17-01752-f004:**
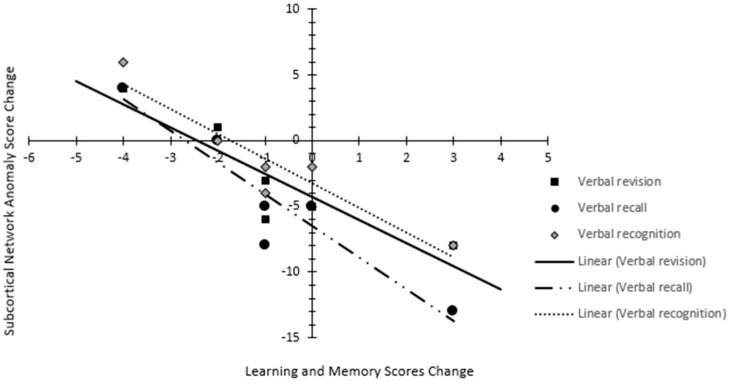
Scatter plot depicting the relation between anomaly change in subcortical network and changes in learning and memory domain (verbal revision, verbal recall, and verbal recognition) score.

**Table 1 cancers-17-01752-t001:** Intricacies of Brain Networks.

S. No.	Brain Network	No. of Parcellations	Network Description	Clinical Significance	Ref.
1	Sensorimotor network (SEN)	30	Encompasses the primary motor, primary sensory, supplementary motor area, dorsal premotor area, ventral premotor area, and cingulate premotor areas.	Complex motor planning and regulation of muscles in the upper limb and upper trunk muscles. Control of hand movements and in performing specific motor tasks based on visual cues.	[30,31]
2	Default mode network (DMN)	27	Encompasses the medial prefrontal cortex, posterior cingulate cortex, precuneus, and lateral/medial temporal lobes.	Dominant network of the task-negative system. Involved in internal mental states, theory of mind, and numerous other cognitive processes. Frequently abnormal in numerous mental illnesses, including Alzheimer’s. Tumors in the left hemisphere and cerebellum are observed to have a significant effect on DMN.	[32,33,34]
3	Central Executive network (CEN)	24	Encompasses dorsolateral prefrontal cortex, posterior parietal cortex.	Dominant network of the task-positive system (active system). CEN is crucial for actively maintaining and manipulating information in working memory, for rule-based problem-solving, and for decision-making in the context of goal-directed behavior.	[35,36,37]
4	Salience network (SAL)	49	The main components are insula and cingulate cortex with 3 key subcortical structures: amygdala, ventral striatum, and substantia nigra/ventral tegmental area.	Plays a role in transitioning between the DMN and CEN, as well as possibly playing a role in pain sensation. Involved in anticipating outcomes, recognizing reward values, and encoding errors to influence attention allocation and motor preparation.	[38,39]
5	Limbic network (LIM)	24	Includes hippocampus, temporal pole, amygdala, posterior cingulate gyrus, and medial, lateral orbitofrontal cortex.	Responsible for conditioned emotional learning, emotional expression, assessment of motivational content, social and emotional processing, and encoding long-term memories.	[40,41]
6	Para limbic network (LIMPA)	34	Includes orbitofrontal cortex, intermodal frontal lobe, and fusiform gyrus.	Exhibits key integrations with the limbic network and plays a crucial role in regulating brain resource allocation; integrates attention, awareness, and emotion.	[42,43]
7	Multiple demand extended network (MDE)	56	Encompasses lateral prefrontal cortex in the inferior frontal sulcus, frontal operculum, intra-parietal sulcus and bilateral pre-supplementary motor area and adjacent middle cingulate cortex (pre-SMA/MCC), anterior insula (aINS), middle frontal gyrus/posterior inferior frontal sulcus/(MFG/IFS).	Plays a role in higher cognitive functions like novelty, perceptual difficulty, conflict response, and different types of memory.	[44]
8	Auditory network (AUD)	30	Encompasses superior temporal gyrus, inferior frontal gyrus, medial frontal gyrus, adjacent insula.	Responsible for discriminating auditory information. Involved in interpreting incoming speech and determining volume, frequency, and onset time of auditory information.	[45,46]
9	Language network (LAN)	15	Includes posterior temporal areas (Wernicke’s area), parietal areas (Geschwind’s area), inferior frontal areas (Broca’s area 44 and 45), supplementary motor area (SMA), area 55b, and area 8C.	Responsible for text comprehension and articulating thoughts into verbal and written words.	[47]
10	Ventral attention network (VAN)	11	Encompasses the lateral and inferior frontal/prefrontal cortex and the temporo-parietal junction.	Involved in attention switching, visual–spatial perception, episodic memory retrieval, and consciousness.	[48,49]
11	Dorsal attention network (DAN)	37	Encompasses the dorsolateral prefrontal cortex, left/right posterior intraparietal sulci, and the frontal eye fields.	Mainly involved in goal-directed attention. Also plays a role in working memory, motor, and executive functions.	[48,49]
12	Subcortical network (SCN)	14	Includes basal ganglia, diencephalon, cerebellum, brain stem, and thalamus.	Plays a regulatory role in cognitive processing, sensory gating, and learning. Implicated in the development of Parkinson’s disease, psychosis, and bipolar disorder.	[50,51,52]
13	Medial temporal network (MT)	16	Encompasses perirhinal ectorhinal cortex (part of temporal lobe), para hippocampal regions PHA1, 2, and 3.	Critically involved in visuospatial processing and episodic memory by processing contextual information. MT-related abnormalities have been extensively demonstrated in patients with temporal lobe epilepsy.	[53]
14	Visual network (VIS)	52	Includes primary visual areas (V1-V4), dorsal visual stream, ventral visual stream, and the lateral visual stream.	Primarily involved in visual processing, and plays role in hand–eye coordination.	[54]
15	Accessory language network (ACLAN)	4	Pathway in the middle temporal gyrus, superior temporal sulcus.	Involved in verbal memory, speech recognition, and the representation of lexical concepts.	[55]

**Table 2 cancers-17-01752-t002:** Details of Neurocognitive Assessment.

Neurocognitive Domain	Test	Description
Learning and memory	Verbal recall	Total number of words correctly recalled over 3 trials
	Verbal revision	Total number of words correctly recalled
	Verbal recognition	Total number of list words correctly identified minus total number of non-list words incorrectly identified
Attention and speed of processing	Symbol match	Total number of correct answers
Executive functions	Ordering numbers and letters	Total time in whole seconds to complete the array
Verbal fluency	Words that start with	Total number of valid words
Fine motor speed	Ordering numbers	Total time in whole seconds to complete the array

**Table 3 cancers-17-01752-t003:** Anomaly frequency changes between baseline and post-WBRT.

S. No.	Brain Network	Baseline Anomaly Frequency	Post-WBRT Anomaly Frequency	Proportional Change (%)
1	Sensorimotor network (SEN)	57	40	−29.82%
2	Default mode network (DMN)	18	8	−55.56%
3	Central Executive network (CEN)	74	82	10.81%
4	Salience network (SAL)	105	101	−3.81%
5	Limbic network (LIM)	30	52	73.33%
6	Para limbic network (LIMPA)	107	161	50.47%
7	Multiple demand extended network (MDE)	258	242	−6.20%
8	Auditory network (AUD)	18	13	−27.78%
9	Language network (LAN)	21	23	9.52%
10	Ventral attention network (VAN)	6	6	0.00
11	Dorsal attention network (DAN)	78	81	3.85%
12	Subcortical network (SCN)	12	15	25.00%
13	Medial temporal network (MT)	7	9	28.57%
14	Visual network (VIS)	126	79	−37.30%
15	Accessory language network (ACLAN)	0	2	-

## Data Availability

Research data are stored in an institutional repository and will be shared upon request to the corresponding author.

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
