# Peer review of "Correlation Between Neurocognitive Outcomes and Neuroaxonal Connectome Alterations After Whole Brain Radiotherapy: A Proof-of-Concept Study†"

_cancers, 2025, doi:10.3390/cancers17111752_

Round 1
Reviewer 1 Report
Comments and Suggestions for Authors
General comments
The study is significant as it is the first to evaluate the impact of whole-brain radiotherapy (WBRT) on the human connectome and investigate the relationship between the observed changes and cognitive function.
To further emphasize the novelty of the research, it would be beneficial to provide a detailed comparison with existing studies (e.g., differences from other radiation therapy research).
Please refer to the attached file for specific comments.
Overall, it is well written and contains interesting findings in the results. However, there are some challenges remaining, such as the small sample size. I believe it would be beneficial to make revisions or additions regarding the points raised in the review.
Methods:
Overall, the study is well-conducted, but the small sample size (n=10) limits the generalizability of the results. While future large-scale cohort studies are anticipated to provide more robust findings, it is important to clearly acknowledge the current limitations of the study.
Line 94: Including detailed information about the machine learning model used in the analysis, such as the algorithms and training data, enhances the reproducibility.
Results:
The results are clearly articulated.
3.1
Line 147: Both of these treatments such as surgery or SRS are likely to have a significant impact on the connectome analysis at baseline prior to WBRT. Therefore, it is important to include information regarding the size of tumor, timing and location of these previous treatments, as the results appear to be associated with them.
Line 149: Additionally, targeted therapy and immunotherapy were administered to some patients after WBRT, both of which are known to affect cerebral metastases, specifically impacting brain tissue. This suggests that these treatments may also influence the connectome analysis following WBRT. At the very least, it is essential to mention the timing of these subsequent treatments.
3.2
It would be beneficial to discuss the networks that exhibited a high proportion of abnormalities in the connectome analysis at baseline prior to WBRT, in relation to the location of the tumors.
3.3
It would also be useful to mention whether the networks that showed a decrease in abnormal frequency are related to changes associated with tumor shrinkage and their positional relationship with the tumors.
Discussion:
The discussion is overall well balanced. The exploration of the effects of radiation therapy on brain networks and the potential for treatment guidelines based on the connectome is intriguing. It is necessary to consider the influence of sample size regarding the lack of statistically significant differences obtained.
Line 210: In the results, it is noted that there are networks with an increased frequency of abnormalities after WBRT and others with a decreased frequency of abnormalities. However, the discussion states that there was no tumor progression. It would be beneficial to consider whether there is a possibility of decreased abnormal frequency associated with tumor shrinkage, as well as the potential correlation with dose, since even with whole-brain radiation therapy, the dose is not necessarily uniform within the brain tissue.
Line 215: The difference in the frequency of abnormalities in subcortical network (SCN) between hippocampal-avoidant WBRT and conventional WBRT is an intriguing finding. It would enhance understanding to discuss whether the SCN is particularly developed in the hippocampus, which is among its key components, including the basal ganglia, thalamus, hypothalamus, and hippocampus.
Line 232: Considering the potential for network improvement associated with tumor shrinkage from radiation, as well as the positional relationship to the tumor, and discussing whether the increase in abnormal frequency is temporary or permanent, would further clarify the clinical significance.

Author Response
We wish to express our appreciation for the in-depth comments and suggestions which have improved the quality of our submitted manuscript. Our responses are in red and bold. We hope the revised version will be acceptable for publication in Cancers.
General comments
The study is significant as it is the first to evaluate the impact of whole-brain radiotherapy (WBRT) on the human connectome and investigate the relationship between the observed changes and cognitive function.
To further emphasize the novelty of the research, it would be beneficial to provide a detailed comparison with existing studies (e.g., differences from other radiation therapy research).
Please refer to the attached file for specific comments.
Overall, it is well written and contains interesting findings in the results. However, there are some challenges remaining, such as the small sample size. I believe it would be beneficial to make revisions or additions regarding the points raised in the review.
We appreciate the reviewer for the positive comments. Although a variety of studies have investigated the relationship between surgery or the impact of other benign processes on the human connectome, the relationship between radiotherapy and connectomic changes has yet to be empirically demonstrated. Therefore, we believe this pilot work is important to add to the literature and encourage further research.
Comment 1
Methods:
Overall, the study is well-conducted, but the small sample size (n=10) limits the generalizability of the results. While future large-scale cohort studies are anticipated to provide more robust findings, it is important to clearly acknowledge the current limitations of the study.
Thank you for the suggestion. We have added “While our current study is limited by its sample size, which may affect the generalizability of results, we plan to investigate the trends in network alterations with reconfigurations over time in a larger cohort with longer follow-up” to the discussion section.
Comment 2: Line 94: Including detailed information about the machine learning model used in the analysis, such as the algorithms and training data, enhances the reproducibility.
Thank you for the suggestion. The brain maps were created by FDA-cleared AI brain mapping software, Quicktome Discovery mode™. The algorithms and training data were detailed in an earlier publication and we provided a citation to the AI mapping information for readers to be able to better understand the algorithms used and the data produced. In brief, the training data from 178 healthy control subjects was obtained from Schizconnect (http://schizconnect.org) and 40 brain tumor patients. A gradient-boosted decision tree model (XGBoost) was used to build a statistical map between the voxel level feature vector and the most probable parcellation class, where “parcellation class” refers to any single brain region included in the A‐HCP atlas.
Doyen S, Nicholas P, Poologaindran A, Crawford L, Young IM, Romero-Garcia R, Sughrue ME. Connectivity-based parcellation of normal and anatomically distorted human cerebral cortex. Hum Brain Mapp. 2022 Mar;43(4):1358-1369. doi: 10.1002/hbm.25728. Epub 2021 Nov 26. PMID: 34826179; PMCID: PMC8837585.
Comment 3:
Results:
The results are clearly articulated.
We thank the reviewer for the positive comments.
Comment 4:
3.1
Line 147: Both of these treatments such as surgery or SRS are likely to have a significant impact on the connectome analysis at baseline prior to WBRT. Therefore, it is important to include information regarding the size of tumor, timing and location of these previous treatments, as the results appear to be associated with them.
Thank you for the suggestion. Among the 2 patients that had prior surgery, one of them also underwent subsequent SRS. So the effect of prior intracranial treatment (n=5) was analyzed together. The median duration between prior intra-cranial treatment and WBRT was 7 months. The patients who received any prior intracranial treatment had a substantial increase in the anomaly frequency of 53%, while those with no prior treatment had a decrease of 33%. Although it is difficult to make conclusions based on these findings alone given the small sample size, we had added this information to the revised manuscript.
Comment 5:
Line 149: Additionally, targeted therapy and immunotherapy were administered to some patients after WBRT, both of which are known to affect cerebral metastases, specifically impacting brain tissue. This suggests that these treatments may also influence the connectome analysis following WBRT. At the very least, it is essential to mention the timing of these subsequent treatments.
Thank you for the suggestion. The median duration between WBRT completion and start of systemic therapy was 2 weeks. The systemic therapy described in the results were between completion of RT and connectome analysis (within 3 months). This is now clarified in the revised results section as well.
Comment 6:
3.2
It would be beneficial to discuss the networks that exhibited a high proportion of abnormalities in the connectome analysis at baseline prior to WBRT, in relation to the location of the tumors.
Thank you for the suggestion. However, in this proof-of-concept study, we focused on the detection of WBRT- induced connectomic changes and their correlation to neurocognitive outcomes. Most of the patients in this study had numerous (more than 15 brain metastases) and diffuse intracranial disease, making analysis in relation to location of tumors in the current study impractical. We plan to evaluate relation to tumor number, location, size, radiation dose in our future studies with a larger sample size.
Comment 7:
3.3
It would also be useful to mention whether the networks that showed a decrease in abnormal frequency are related to changes associated with tumor shrinkage and their positional relationship with the tumors.
As mentioned above, the mean number of brain metastases in this study was 35 with some patients with approximately 80-100 lesions. Therefore, we were unable to characterize the impact of tumor volume changes and position on connectomic changes. In a separately planned study, we aim to better evaluate this with patients with limited brain metastasis undergoing SRS. But, for this project, we aimed to evaluate the global changes that followed treatment to the entirety of the brain.
Discussion:
The discussion is overall well balanced. The exploration of the effects of radiation therapy on brain networks and the potential for treatment guidelines based on the connectome is intriguing. It is necessary to consider the influence of sample size regarding the lack of statistically significant differences obtained.
Comment 8:
Line 210: In the results, it is noted that there are networks with an increased frequency of abnormalities after WBRT and others with a decreased frequency of abnormalities. However, the discussion states that there was no tumor progression. It would be beneficial to consider whether there is a possibility of decreased abnormal frequency associated with tumor shrinkage, as well as the potential correlation with dose, since even with whole-brain radiation therapy, the dose is not necessarily uniform within the brain tissue.
Thank you for the suggestion. At the 3-month follow up imaging, we observed that none of the patients had increase in number or size of metastases. While the decrease in anomaly frequency in some networks could be due to tumor shrinkage, since most of them have a substantial intracranial disease burden (mean 35 metastases, with some patients with 80-100 lesions), it is impossible to characterize this in the current study given the limited sample size alone. Patients with 5 or fewer brain lesions receiving SRS might be ideal candidates to study such correlations. Incorporating radiation dose constitutes the next phase in an attempt to yield connectome-tailored treatment directive.
Comment 9:
Line 215: The difference in the frequency of abnormalities in subcortical network (SCN) between hippocampal-avoidant WBRT and conventional WBRT is an intriguing finding. It would enhance understanding to discuss whether the SCN is particularly developed in the hippocampus, which is among its key components, including the basal ganglia, thalamus, hypothalamus, and hippocampus.
Thank you for the suggestion. We have added an additional paragraph in the discussion to expand on this hypothesis.
“The key components of the SCN include the intricate connections between the basal ganglia, thalamus, hypothalamus, and hippocampus. Traditionally, these structures were studied as isolated anatomic loci, however, their reconceptualization as networks with coordinated neuronal activity and dynamic hubs contributing to critical cognitive functions has resulted in significant clinical implications. Learning and memory processes rely extensively on the dynamic interaction between cortical and subcortical structures, with differential activity across memory encoding, consolidation, and retrieval phases [58,59]. Thalamic sub-nuclei are well known to form functional circuits with extensive reciprocal connections to cortical areas, and act as a dynamic filter of information flow. The mediodorsal nucleus enhances fronto-parietal interactions, stabilizing information during the “verbal recall” phase [60]. Concurrently, the basal ganglia reinforces verbal associations via dopaminergic signaling and is central to procedural learning [61]. Thus, “verbal revision” and “verbal recognition” require coordinated thalamocortical and basal ganglia-cortical interactions. Xu et al. compared individuals with subjective cognitive decline to healthy controls and reported significant structural and functional alterations in cortical-subcortical circuit and disappearance of some hubs [62]. The hippocampus binds multimodal sensory, temporal and contextual details into coherent memories. It interacts extensively with medial prefrontal cortex (mPFC) and plays a key role in episodic and spatial memory [63]. Dynamic causal modeling showed effective connectivity between the hippocampus and DMN during encoding and memory retrieval [64]. Although a statistical difference is not attained due to the small sample size of the current study, the difference observed in the change in SCN anomaly frequency between hippocampal avoidant and conventional WBRT reinforces the effect of RT on brain networks and might indicate a threshold dose responsible for anomalies that attain clinical significance.”
Comment 10:
Line 232: Considering the potential for network improvement associated with tumor shrinkage from radiation, as well as the positional relationship to the tumor, and discussing whether the increase in abnormal frequency is temporary or permanent, would further clarify the clinical significance.
We are following up patients with subsequent scans at 6 months to assess the nature of radiation-induced changes. However, the necessary data is forthcoming.

Reviewer 2 Report
Comments and Suggestions for Authors
Review of "Correlation between Neurocognitive Outcomes and Neuroaxonal Connectome Alterations after Whole Brain Radiotherapy: A Proof-of-Concept Study"
In this prospective pilot study, the authors explore the impact of whole brain radiotherapy (WBRT) on the brain’s structural and functional connectivity using diffusion tensor imaging (DTI) and functional MRI (fMRI). By applying a machine learning algorithm to create individualized brain maps and performing comprehensive neurocognitive testing, they aim to correlate connectomic alterations with neurocognitive outcomes. Their results suggest a significant association between changes in brain network integrity and cognitive decline, particularly within the subcortical network. This proof-of-concept study provides intriguing preliminary evidence supporting the clinical relevance of connectomic analysis in patients undergoing WBRT.
Methods
The authors employed a novel, unvalidated neurocognitive test platform. While this innovative approach is intriguing, it introduces uncertainty regarding the validity of the results. I would encourage the authors to elaborate on how their testing platform correlates with commonly used, validated instruments such as the Boston Naming Test (BNT) or the Rey Auditory Verbal Learning Test Long Delay (RAVLT-LD). Was any formal validation performed? If so, please provide details.
The definitions of hyperactivation and hypoactivation require clarification. How were these terms operationalized within the anomaly analysis? Were anomalies analyzed as dichotomous variables (hyper- vs. hypoactivation), or as continuous values based on thresholds (e.g., > |0.8|)? Further, while hyperactivation and hypoactivation were grouped together under the umbrella of "anomalies," the potential clinical differences between these two phenomena should be discussed. If they are biologically or clinically distinct, the rationale for grouping them should be justified.
The method for calculating anomaly frequency is unclear. When the authors report that "the highest proportion of anomalies per network was noted in the multiple demand network (MDE: 46%)," does this 46% represent the average anomaly frequency across all patients, or is it derived differently? Please clarify the approach for better reproducibility.
Additionally, the criteria for defining neurocognitive decline need further explanation. The authors state that decline was defined by “a change of point score in each tested domain.” Does this imply that even a single point change constituted a decline? Was a cutoff range employed? It would strengthen the study to consider Z-scores or other standardized methods for defining significant decline.
Results
In Figure 3, the authors describe decreased fiber density in structural connectivity at baseline, with further deterioration post-WBRT. However, the methods section does not detail a separate, independent analysis of structural connectivity. How was decreased fiber density determined—through subjective visual assessment or through quantitative measures? If subjective, please specify, and consider discussing the potential limitations related to software variability or reconstruction artifacts.
Moreover, without a direct comparison group or internal software validation, how can the authors be certain that observed changes in fiber tracts post-WBRT are due to treatment effects rather than technical variation in reconstruction? Addressing this point would enhance the rigor of the analysis.
Regarding Figure 4, please clarify within the figure legend that "SCN" refers to the Subcortical Network for easier interpretation.
Finally, while the authors found correlations between cognitive decline and changes in the subcortical network, it is unclear why learning and memory outcomes—typically associated with limbic and paralimbic structures—were correlated specifically with the subcortical network. Clarification of this point, including possible neuroanatomical or functional explanations, would strengthen the interpretation of the results.
Author Response
We wish to express our appreciation for the in-depth comments and suggestions which have improved the quality of our submitted manuscript. Our responses are in red and bold. We hope the revised version will be acceptable for publication in Cancers.
In this prospective pilot study, the authors explore the impact of whole brain radiotherapy (WBRT) on the brain’s structural and functional connectivity using diffusion tensor imaging (DTI) and functional MRI (fMRI). By applying a machine learning algorithm to create individualized brain maps and performing comprehensive neurocognitive testing, they aim to correlate connectomic alterations with neurocognitive outcomes. Their results suggest a significant association between changes in brain network integrity and cognitive decline, particularly within the subcortical network. This proof-of-concept study provides intriguing preliminary evidence supporting the clinical relevance of connectomic analysis in patients undergoing WBRT.
Comment 1:
Methods
The authors employed a novel, unvalidated neurocognitive test platform. While this innovative approach is intriguing, it introduces uncertainty regarding the validity of the results. I would encourage the authors to elaborate on how their testing platform correlates with commonly used, validated instruments such as the Boston Naming Test (BNT) or the Rey Auditory Verbal Learning Test Long Delay (RAVLT-LD). Was any formal validation performed? If so, please provide details.
Response 1: We do agree with the reviewer that the neurocognitive test platform is a novel test. Leemans et al. assessed this tool in comparison to well-established paper and pencil tests and demonstrated satisfactory reliability. We have used this multi-dimensional testing program in over 250 patients at our institution, and a recent study in the brain metastasis patients demonstrated the feasibility and its ability to discriminate neurocognitive decline in patients with brain metastases treated with WBRT and SRS. These are added to the revised version of the text. We have added this as a limitation to the current work as well.
Leemans, K.; De Ridder, M. Cognition: Development of a Cognitive Testing Battery on the iPad for the Evaluation of Patients with Brain Mets. Acta Neurol Belg 2022, 122, 145–152, doi:10.1007/s13760-021-01744-9.
Akdemir, E.Y.; Gurdikyan, S.; Reyes, T.C.; Odia, Y.; Menendez, M.A.R.; Yarlagadda, S.; Gal, O.; Hall, M.D.; Press, R.H.; Wieczorek, D.J.; et al. Integrating a Novel Tablet-Based Digital Neurocognitive Assessment Tool in Brain Metastases Patients. J Neurooncol 2025, doi:10.1007/s11060-025-05038-5.
Comment 2: The definitions of hyperactivation and hypoactivation require clarification. How were these terms operationalized within the anomaly analysis? Were anomalies analyzed as dichotomous variables (hyper- vs. hypoactivation), or as continuous values based on thresholds (e.g., > |0.8|)? Further, while hyperactivation and hypoactivation were grouped together under the umbrella of "anomalies," the potential clinical differences between these two phenomena should be discussed. If they are biologically or clinically distinct, the rationale for grouping them should be justified.
Response 2: The brain maps were compared to the normal brain from the Human connectome project and a 3 standard deviation alteration from the normal brain was defined as an “anomaly”. Hyperactivation was defined as the area generating more neuronal activity compared to normal, and hypoactivation when showing depressed neuronal activity. This is now clarified in the revised version of the text. These were grouped together for the purpose of the current analysis, given the preliminary nature, the sample size limits the statistical power to characterize them accurately.
Comment 3: The method for calculating anomaly frequency is unclear. When the authors report that "the highest proportion of anomalies per network was noted in the multiple demand network (MDE: 46%)," does this 46% represent the average anomaly frequency across all patients, or is it derived differently? Please clarify the approach for better reproducibility.
Response 3: Thank you for the suggestion. This was computed as an average across all patients and has been added to the text now.
Comment 4: Additionally, the criteria for defining neurocognitive decline need further explanation. The authors state that decline was defined by “a change of point score in each tested domain.” Does this imply that even a single point change constituted a decline? Was a cutoff range employed? It would strengthen the study to consider Z-scores or other standardized methods for defining significant decline.
Response 4: Thank you for the suggestion. “Neurocognitive function change was computed using Iverson modification of Reliable Change Index (RCI), and in accordance with the previous literature, a z-score of ±1.645 (90% confidence level) was defined as reliable change.”
Duff K. Evidence-based indicators of neuropsychological change in the individual patient: relevant concepts and methods. Arch Clin Neuropsychol. 2012 May;27(3):248-61. doi: 10.1093/arclin/acr120. Epub 2012 Feb 29. PMID: 22382384; PMCID: PMC3499091.
Comment 5:
Results
In Figure 3, the authors describe decreased fiber density in structural connectivity at baseline, with further deterioration post-WBRT. However, the methods section does not detail a separate, independent analysis of structural connectivity. How was decreased fiber density determined—through subjective visual assessment or through quantitative measures? If subjective, please specify, and consider discussing the potential limitations related to software variability or reconstruction artifacts.
Response 5: Figure 3 is only a single representative case showing a single network that showcases the visualized changes on the baseline and post-WBRT scans. The machine learning algorithm integrates data from DTI and fMRI to analyze structural and functional connectivity together and creates the networks. Structural connectivity was not otherwise formally analyzed.
Comment 6: Moreover, without a direct comparison group or internal software validation, how can the authors be certain that observed changes in fiber tracts post-WBRT are due to treatment effects rather than technical variation in reconstruction? Addressing this point would enhance the rigor of the analysis.
Response 6: The networks were created by an FDA-cleared AI brain mapping software. The image capturing technique and processing software were the same for baseline and post-WBRT scans to ensure uniformity. Hence, the observed changes are associated with the intervening intervention (WBRT). This is now clarified in the text.
Comment 7: Regarding Figure 4, please clarify within the figure legend that "SCN" refers to the Subcortical Network for easier interpretation.
Response 7: Thank you for the suggestion. Figure 4 is now replaced as advised.
Comment 8: Finally, while the authors found correlations between cognitive decline and changes in the subcortical network, it is unclear why learning and memory outcomes—typically associated with limbic and paralimbic structures—were correlated specifically with the subcortical network. Clarification of this point, including possible neuroanatomical or functional explanations, would strengthen the interpretation of the results.
Response 8: Thank you for the suggestion. An additional paragraph is added to the discussion detailing this.
“The key components of the SCN include the intricate connections between the basal ganglia, thalamus, hypothalamus, and hippocampus. Traditionally, these structures were studied as isolated anatomic loci, however, their reconceptualization as networks with coordinated neuronal activity and dynamic hubs contributing to critical cognitive functions has resulted in significant clinical implications. Learning and memory processes rely extensively on the dynamic interaction between cortical and subcortical structures, with differential activity across memory encoding, consolidation, and retrieval phases [58,59]. Thalamic sub-nuclei are well known to form functional circuits with extensive reciprocal connections to cortical areas, and act as a dynamic filter of information flow. The mediodorsal nucleus enhances fronto-parietal interactions, stabilizing information during the “verbal recall” phase [60]. Concurrently, the basal ganglia reinforces verbal associations via dopaminergic signaling and is central to procedural learning [61]. Thus, “verbal revision” and “verbal recognition” require coordinated thalamocortical and basal ganglia-cortical interactions. Xu et al. compared individuals with subjective cognitive decline to healthy controls and reported significant structural and functional alterations in cortical-subcortical circuit and disappearance of some hubs [62]. The hippocampus binds multimodal sensory, temporal and contextual details into coherent memories. It interacts extensively with medial prefrontal cortex (mPFC) and plays a key role in episodic and spatial memory [63]. Dynamic causal modeling showed effective connectivity between the hippocampus and DMN during encoding and memory retrieval [64]. Although a statistical difference is not attained due to the small sample size of the current study, the difference observed in the change in SCN anomaly frequency between hippocampal avoidant and conventional WBRT reinforces the effect of RT on brain networks and might indicate a threshold dose responsible for anomalies that attain clinical significance.”

Reviewer 3 Report
Comments and Suggestions for Authors
The paper is about evaluating the effect of whole brain radiotherapy on the connectome. The authors combine diffusion tensor imaging and functional magnetic resonance imaging to study the functional connectivity of the brain. They find that brain connectivity changes with WBRT is correlated with neurocognitive outcomes. The paper is interesting, but its presentation needs minor improvements.
(1) in keywords, it is better to not to use abbreviations as keywords. Magnetic resonance overlaps with fMRI.
(2) from page 4 to page 10, table 1 is badly organized. Table has to be concise and easy to understand.
(3) in abstract, “a machine learning algorithm trained to analyze ”, what machine learning algorithm is used in the paper. The authors have to make a explanation about which algorithm and how it is used in the paper.
(4) the authors say “this pilot study integrated data from DTI and fMRI”. A simple search gives two existing papers combined DTI and fMRI as follows. Hence, the combination is not a pilot study. Please check the conclusion.
Fernández-Espejo, D., Junque, C., Cruse, D. et al. Combination of diffusion tensor and functional magnetic resonance imaging during recovery from the vegetative state. BMC Neurol 10, 77 (2010). doi: 10.1186/1471-2377-10-77.
Kokkinos V, Chatzisotiriou A, Seimenis I. Functional Magnetic Resonance Imaging and Diffusion Tensor Imaging-Tractography in Resective Brain Surgery: Lesion Coverage Strategies and Patient Outcomes. Brain Sci. 2023 Nov 9;13(11):1574. doi: 10.3390/brainsci13111574.
(5) the “increase” conclusion should be explained in detail in section 3.2, 3.3 and 3.4. it is better to present the results in a table. The table shows which network increases or not increases, and the ratio of increase.
Author Response
We wish to express our appreciation for the in-depth comments and suggestions which have improved the quality of our submitted manuscript. Our responses are in red and bold. We hope the revised version will be acceptable for publication in Cancers.
The paper is about evaluating the effect of whole brain radiotherapy on the connectome. The authors combine diffusion tensor imaging and functional magnetic resonance imaging to study the functional connectivity of the brain. They find that brain connectivity changes with WBRT is correlated with neurocognitive outcomes. The paper is interesting, but its presentation needs minor improvements.
Comment 1: in keywords, it is better to not to use abbreviations as keywords. Magnetic resonance overlaps with fMRI.
Response 1: Thank you for the suggestion. This has now been updated.
Comment 2: from page 4 to page 10, table 1 is badly organized. Table has to be concise and easy to understand.
Response 2: Thank you for the suggestion. The table has now been re-formatted as advised. Since this table gives a comprehensive overview of all the networks employed in the current study, this was longer than expected.
Comment 3: in abstract, “a machine learning algorithm trained to analyze ”, what machine learning algorithm is used in the paper. The authors have to make a explanation about which algorithm and how it is used in the paper.
Response 3: The brain networks were created by a FDA-cleared AI brain mapping software. This is now clarified in the text.
Comment 4: the authors say “this pilot study integrated data from DTI and fMRI”. A simple search gives two existing papers combined DTI and fMRI as follows. Hence, the combination is not a pilot study. Please check the conclusion.
Fernández-Espejo, D., Junque, C., Cruse, D. et al. Combination of diffusion tensor and functional magnetic resonance imaging during recovery from the vegetative state. BMC Neurol 10, 77 (2010). doi: 10.1186/1471-2377-10-77.
Kokkinos V, Chatzisotiriou A, Seimenis I. Functional Magnetic Resonance Imaging and Diffusion Tensor Imaging-Tractography in Resective Brain Surgery: Lesion Coverage Strategies and Patient Outcomes. Brain Sci. 2023 Nov 9;13(11):1574. doi: 10.3390/brainsci13111574.
Response 4: We changed the wording in the conclusion as “proof-of-concept study”. This is the first analysis evaluating the whole-brain radiation induced connectome changes.
Comment 5: the “increase” conclusion should be explained in detail in section 3.2, 3.3 and 3.4. it is better to present the results in a table. The table shows which network increases or not increases, and the ratio of increase.
Response 5: Thank you for the suggestion. Table 3 has now been added as suggested.

Round 2
Reviewer 2 Report
Comments and Suggestions for Authors
The revised version fully addresses all of the reviewer's comments.
Reviewer 3 Report
Comments and Suggestions for Authors
It is bad to upload a manuscript with tracked changes in the document. It is hard to read. Some sentences have grammar errors. Overall, the paper has been improved and it is acceptable after checking grammar errors. For example, on page 4, "was chosen" appears twice.